# Secondary Metabolites from the Mangrove Ecosystem-Derived Fungi *Penicillium* spp.: Chemical Diversity and Biological Activity

**DOI:** 10.3390/md23010007

**Published:** 2024-12-26

**Authors:** Guojun Zhou, Jin Cai, Bin Wang, Wenjiao Diao, Yu Zhong, Shaodan Pan, Weijia Xiong, Guolei Huang, Caijuan Zheng

**Affiliations:** 1Key Laboratory of Tropical Medicinal Resource Chemistry of Ministry of Education, College of Chemistry and Chemical Engineering, Hainan Normal University, Haikou 571158, China; yaoxuewsw@163.com (G.Z.); caijin20210207@163.com (J.C.); 15180328507@163.com (B.W.); diaowj@qibebt.ac.cn (W.D.); washaixind@163.com (Y.Z.); 18084603981@163.com (S.P.); 2Key Laboratory of Tropical Medicinal Plant Chemistry of Hainan Province, Haikou 571158, China; 3Yunnan Key Laboratory of Screening and Research on Anti-Pathogenic Plant Resources from Western Yunnan, Institute of Materia Medica, College of Pharmacy, Dali University, Dali 671000, China; 13577544542@163.com

**Keywords:** mangrove ecosystems, *Penicillium* sp., secondary metabolites, chemical diversity, biological activity

## Abstract

Mangrove ecosystems have attracted widespread attention because of their high salinity, muddy or sandy soil, and low pH, as well as being partly anoxic and periodically soaked by tides. Mangrove plants, soil, or sediment-derived fungi, especially the *Penicillium* species, possess unique metabolic pathways to produce secondary metabolites with novel structures and potent biological activities. This paper reviews the structural diversity and biological activity of secondary metabolites isolated from mangrove ecosystem-derived *Penicillium* species over the past 5 years (January 2020–October 2024), and 417 natural products (including 170 new compounds, among which 32 new compounds were separated under the guidance of molecular networking and the OSMAC approach) are described. The structures were divided into six major categories, including alkaloids, polyketides, terpenoids, benzene derivatives, steroids, and other classes. Among these natural products, the plausible biosynthetic pathways of 37 compounds were also proposed; 11 compounds have novel skeleton structures, and 26 compounds contain halogen atoms. A total of 126 compounds showed biological activities, such as cytotoxic, antifungal, antibacterial, anti-inflammatory, and *α*-glucosidase-inhibitory activities, and 11 compounds exhibited diverse biological activities. These new secondary metabolites with novel structures and potent bioactivities will continue to guide the separation or synthesis of structurally novel and biologically active compounds and will offer leading compounds for the development and innovation of pharmaceuticals and pesticides.

## 1. Introduction

Nowadays, the spread of drug-resistant bacteria, frequent cancer, and crop diseases all affect human life and health. With respect to resistant bacteria, antibiotic resistance in methicillin-resistant *Staphylococcus aureus* (MRSA) and *Pseudomonas aeruginosa* remains one of the most challenging phenomena of everyday medical science. The *β*-lactamase (Bla) secreted by MRSA hydrolyzes nearly all *β*-lactam antibiotics, leaving only a few antibiotics available for the clinical treatment of MRSA infections, which are among the leading pathogens for death [1]. The adaptive antibiotic resistance of *P. aeruginosa*, including biofilm-mediated resistance and the formation of multidrug-tolerant persisted cells, is accountable for the recalcitrance and death of infections [2]. Meanwhile, some emerging multidrug-resistant pathogens, for example, *Candida vulturna* and *Candida auris*, are emerging at an alarming rate and posing serious threats to public health [3,4]. As far as cancer issues are concerned, cancer causes one out of every six deaths around the globe [5]. For example, lung cancer is one of the most frequently diagnosed cancers and the leading cause of cancer-related deaths worldwide, with an estimated 2 million new cases and 1.76 million deaths per year [6]. As for agricultural production, the worldwide reduction in food production due to pests and diseases is still an important challenge [7,8]. Therefore, the development of effective leading drug compounds remains an urgent need for the pharmaceutical and pesticide industries in relation to people’s lives and health.

Since 2014, Blunt J.W., Carroll A.R., and their colleagues have highlighted the distinction between mangrove-associated fungi and marine fungi, driven by the proliferation of reported compounds originating from fungi inhabiting mangrove plants and soil [9,10,11,12,13,14,15,16,17,18,19]. This distinction emphasizes the uniqueness and potential of mangrove-derived microorganisms. Based on the above results, more and more researchers have obtained a large number of compounds from mangrove-associated microorganisms [20,21,22,23]. In particular, as essential mangrove-associated organisms, *Penicillium* fungi have received considerable attention among all mangrove-derived fungi, accounting for 28.32% of natural products of mangrove fungal origin, and play an important role in the discovery of natural marine products with bioactivities and novel structures [24,25,26,27]. At the same time, many researchers have reviewed mangrove-associated microorganisms’ natural products from different perspectives. She et al. reviewed discoveries relating to the source and characteristics of metabolic products isolated from mangrove-associated fungi (1989–2020) [27]. Braga et al. reviewed the antibacterial, antifungal, and antiviral compounds produced by soil and sediment-derived mangrove fungi from 1990 to 2022 [27]. Luo et al. summarized 110 compounds and anti-tumor secondary metabolites originating from fungi within the mangrove ecosystems of the South China Sea between 2016 and 2022 [28]. Yu et al. provided an overview of new secondary metabolites from mangrove-associated strains from January 2021 to May 2024 [29]. Gao et al. summarized cytotoxic compounds from marine fungi, detailing their sources, structures, and bioactivity (1991–2023) [30]. Lv et al. focused on novel bioactive natural products from marine-derived *Penicillium* fungi (2021–2023) [31]. Lastly, Li et al. presented a comprehensive overview of 519 natural products isolated from mangrove sediment-derived microbes, along with their bioactivities, up to 2021 [32]. Collectively, these articles and reviews indicate that mangrove ecosystem-derived microorganisms, especially *Penicillium* fungi, may hold significant potential in addressing the three major challenges of drug-resistant bacterial spread, frequent cancer, and crop diseases.

In recent years, we have been focused on the exploration of secondary metabolites derived from *Penicillium* species associated with mangrove ecosystems [33,34,35,36,37,38,39,40,41,42]. In this review, continuing our previous work [43], we update the chemical diversity and biological activity of secondary metabolites produced by *Penicillium* species derived from mangrove plants, soil, or sediment. In particular, the combination of the One Strain Many Compounds (OSMAC) strategy, gene cluster information, and the application of MS/MS data and/or Global Natural Products Social (GNPS)-based molecular networking was used to facilitate the targeted extraction of new compounds. Additionally, the biosynthetic pathways of the isolated compounds were further analyzed. A total of 417 secondary metabolites were obtained from mangrove ecosystem-derived *Penicillium* fungi. Among them, 170 were identified as new compounds, and 126 exhibited bioactivity. These compounds were structurally categorized into alkaloids (121 compounds), polyketides (146 compounds), terpenoids (84 compounds), benzene derivatives (38 compounds), steroids (8 compounds), and other compounds (20 compounds). The structures and absolute configurations of new compounds and novel skeleton compounds were elucidated by detailed spectroscopic analyses, including nuclear magnetic resonance (NMR) spectra, mass spectrometry (MS) data, electronic circular dichroism (ECD) calculations, and single-crystal X-ray diffraction. By examining the latest advancements and trends in this field, our aim is to provide an updated overview of the current research status and technical methods, striving to discover active compounds with medical or agricultural value.

## 2. Structural and Biological Activity Studies

### 2.1. Alkaloids

In general, alkaloids are nitrogen-containing organic compounds (proteins, amino acids, peptides, and vitamin B are not included) with significant biological activity. A total of 124 alkaloids compounds were discovered from the genus *Penicillium* sp., including diketopiperazines (24 compounds), indole diterpenoid alkaloids (34 compounds), quinolinones, isoquinolines and quinolones alkaloids (20 compounds), sclerotioramines (13 compounds), pyridine derivatives (4 compounds), benzodiazepines (6 compounds), and other alkaloid compounds (20 compounds), among which 43 compounds were new compounds and 32 compounds were bioactive compounds.

#### 2.1.1. Diketopiperazines

Eight new dioxopiperazine alkaloids, designated as penispirozines A–H (**1**–**8**), were discovered from the mangrove *Sonneratia caseolaris*-derived fungus *Penicillium janthinellum* HDN13-309 (Hainan province, China). Compound **1** had an unusual pyrazino[1,2]oxazadecaline coupled with a thiophane ring system, and compound **2** possessed a 6/5/6/5/6 pentacyclic ring system with two rare spirocyclic centers. Compounds **3** and **4** could increase the expression of the two relevant phase II detoxifying enzymes SOD2 and HO-1 in HaCaT cells at a concentration of 10 µM [44]. Three known compounds, brevianamide F (**9**), maculosin (**10**), and okaramine U (**11**), were obtained from the mangrove rhizosphere soil of *Bruguiera gymnorrhiza*-derived fungus *Penicillium* sp. DM27 (Chantaburi province, Thailand) [45]. One known compound, cyclo(trp-phe) (**12**), was isolated from *Penicillium polonicum* MCCC3A00951, and the fungus MCCC3A00951 was collected from the mangrove forest of Zhangjiangkou in Fujian province, China [46]. One known compound, (3*E*)-3-[(1*H*-imidazol-4-yl)methylidene]-6-[(1*H*-indol-3-yl)methyl]piperazine-2,5-dione (**13**), was isolated from mangrove rhizosphere soil of *Hibiscus tiliaceus* derived fungus *Penicillium* sp. DM815 (Hainan province, China) [47]. Two known compounds, fructigenine A (**14**) and aurantiomide C (**15**), were identified from the fungus *Penicillium polonicum* H175, which was isolated from sediment of the mangrove national nature reserve in Zhangjiangkou, Fujian province, China [48]. A new trithiodiketopiperazine derivative, adametizine C (**16**), together with five known dithiodiketopiperazine derivatives, adametizine A (**17**), DC1149B (**18**), outovirin B (**19**), pretrichodermamide E (**20**), and peniciadametizine A (**21**), were isolated from the mangrove sediment-derived fungus *Penicillium ludwigii* SCSIO 41408 (Hongsha River estuary of South China Sea, Sanya, Hainan province, China). Compounds **16**–**20** exhibited inhibitory activity for *Erysipelothrix rhusiopathiae* WH13013, with MIC values of 50, 100, 50, 100, and 50 μg/mL, respectively. Compounds **16**, **18**–**20** showed antibacterial activity against *Streptococcus suis* SC19, with MIC values of 100, 50, 100, and 100 μg/mL, respectively. Compounds **16**–**18** showed cytotoxicity against the prostate cancer cell line 22Rv1 (androgen receptor positive), with IC_50_ values of 13.9, 13.0, and 13.6 µM, respectively. Moreover, compound **18** showed potent activity against another prostate cancer PC-3 (androgen receptor negative) cells, with an IC_50_ value of 5.1 µM. Further experiments revealed that **18** could significantly reduce PC-3 cell colony formation and induce apoptosis in a dose-dependent manner. Compounds **16**, **17**, and **20** exhibited obvious inhibitory activities against LPS-induced NF-*κ*B, with IC_50_ values of 8.2, 15.1 and 12.6 µM, respectively [49]. Two known compounds, bisdethiobis(methylthio)-acetylapoaranotin (**22**) and alternarosin A (**23**), were obtained from the mangrove *Acanthus ilicifolius* endophytic fungus *Penicillium* sp. HS-N-29 (South China Sea) [50]. One known compound, fumitremorgin A (**24**), was isolated from mangrove sediment-derived fungus *Penicillium brefeldianum* ABC190807 (Dongzhaigang Mangrove Nature Reserve, Hainan province, China). Compound **24** showed larvicidal activity, with an LC_50_ value of 0.337 mg/mL [51] (Figure 1).

#### 2.1.2. Indole-Diterpenoid Alkaloids

A new prenylated indole alkaloid paraherquamide J (**25**), together with three known compounds, paraherquamide K (**26**), paraherquamide A (**27**), and paraherquamide E (**28**), were isolated from the mangrove rhizosphere soil-derived fungus *Penicillium janthinellum* HK1-6 (Dongzhaigang, Hainan province, China) [52]. Continuing with this research, a new prenylated indole alkaloid, notoamide X (**29**), was isolated from the fungus *P. janthinellum* HK1-6 by employing molecular networking strategies [35]. Seven indole diterpenes, paspaline (**30**), 3-deoxo-4b-deoxypaxilline (**31**), 6,7-dehydropaxilline (**32**), emindole SB (**33**), paspalinine (**34**), paspalitrem C (**35**), and paspalitrem A (**36**), were obtained from the mangrove rhizosphere soil-derived fungus *Penicillium javanicum* HK1-23 (Dongzhaigang, Hainan province, China). Compound **33** had inhibitory activity against *Staphylococcus aureus* ATCC 33591, with an MIC value of 6.25 μg/mL. Compounds **31**, **33**, and **34** exhibited lethality (*Artemia salina*) with LC_50_ values of 3.12, 15.9, and 2.20 μg/mL, respectively [53]. An unprecedented disecoindole diterpenoid peniditerpenoid A (**37**) and a rare N-oxide-containing indole diterpenoid derivative peniditerpenoid B (**38**) were obtained from the mangrove sediment-derived fungus *Penicillium* sp. SCSIO 41411 (Gaoqiao Mangrove in Zhanjiang, Guangdong province, China). Compound **37** exhibited inhibitory effects on lipopolysaccharide-induced NF-*κ*B, with an IC_50_ value of 11 μM. Moreover, **37** effectively prevented RANKL-induced osteoclast differentiation in bone marrow macrophages. In vitro studies demonstrated that **37** exerted significant inhibition of NF-*κ*B activation in the classical pathway by preventing TAK1 activation, I*κ*B*α* phosphorylation, and p65 translocation. Furthermore, **37** effectively reduced the level of NFATc1 activation, resulting in the attenuation of osteoclast differentiation. Compound **37** exhibits considerable promise as an inhibitor with significant potential for the treatment of diseases associated with osteoporosis [54]. Four new indole diterpene analogs, shearinine R (**39**), shearinine S (**40**), 22-hydroxyshearinine I (**41**), shearinine T (**42**), and ten known indole diterpene analogs, 22,23-dehydro-shearinineA (**43**), shearinine A (**44**), shearinines D–G (**45**–**48**), 22-hydroxylshearinine F (**49**), shearinine H (**50**), paspalitrem C (**51**), and dehydroxypaxilline (**52**), were discovered from the rhizosphere soil of the mangrove *Kandelia candel*-derived fungus *Penicillium* sp. UJNMF0740 (Dongzhai Harbor, Hainan province, China). Compounds **44** and **47** exhibited weak growth inhibitory effects against *Staphylococcus aureus* TCCC 25923, with IC_50_ values of 48.4 and 47.6 μM, respectively. Compound **43** showed protective effects against the injury of PC12 cells induced by 6-hydroxydopamine (6-OHDA), and displayed the best ameliorating effect against the damage of PC12 cells at 25 µM. Additionally, compound **43** could suppress the apoptosis and production of reactive oxygen species (ROS) in 6-OHDA-stimulated PC12, and simultaneously induce the phosphorylation of PI3K and Akt. A plausible biosynthetic pathway for **39**–**52** was postulated, whereby compound **51** and shearinine K could be produced through oxidation at C-13, as well as further prenylation and diprenylation on the aromatic ring of **52**, respectively. Subsequently, oxidation and dehydration cyclization on the isopentenyls of shearinine K would afford **47**, while the further oxidation at C-22 of **47** would easily proceed to generate **49** and **48**. The double-bond isomerization from Δ^23,28^ of **47** to Δ^27,28^ would lead to **44**, while the following oxidation and methylation occurring at C-22 of **44** would afford **45**, **42**, and **46**. The oxidation of the respective intermediate (i) by the cleavage of the C-2-C-18 bond could produce an eight-membered ring fragment in **50**. Then, the double-bond isomerization, like from **47** to **44**, and the oxidation at C-22 in **50** would form **41**, and **43** would be derived from the dehydration of **45**. Finally, compound **50** could undergo the hydrolysis of the amide bond to yield **40**, and keto-enol tautomerization and the methylation of **50** would produce **39** (Figure 1) [55]. One undescribed alkaloid, penicioxa A (**53**), and one known compound, oxaline (**54**), were isolated from *Penicillium robsamsonii* HNNU0006, which was isolated from the rhizospheric soil of the mangrove *Sonneratia paracaseolaris* (Dongzhai Gang, Hainan province, China) [36]. Two known compounds, meleagrin (**55**) and neooxaline (**56**), were obtained from the mangrove rhizosphere soil of *Bruguiera gymnorrhiza* (L.) Poir.-derived fungus *Penicillium* sp. DM27 (Chantaburi province, Thailand) [34]. Two known compounds, peniciherquamide A (**57**) and preparaherquamide (**58**), were isolated from mangrove sediment-derived fungus *Penicillium chrysogenum* ZZ1151 (Indonesia) [56] (Figure 2).

#### 2.1.3. Quinolinone, Isoquinoline and Quinolone Alkaloids

One new quinolinone derivative, quinolactone A (**59**), a pair of epimers quinolactacin C1 (**60**) and 3-*epi*-quinolactacin C1 (**61**), together with three known analogs quinolactacins (A2 (**62**), B2 (**63**), and D (**64**)) were isolated from mangrove *Thespesia populnea* endophytic fungus *Penicillium citrinum* YX-002 (Zhanjiang Mangrove Nature Reserve, China). Compounds **59** and **62** showed moderate acetylcholinesterase (AChE) inhibitory activity, with IC_50_ values of 27.6 and 19.4 μM, respectively [57]. A new tetrahydroisoquinoline, peniciisoquinoline A (**65**), was obtained from mangrove rhizosphere soil of *Bruguiera gymnorrhiza* (L.) Poir.-derived fungus *Penicillium* sp. DM27 (Chantaburi province, Thailand) [45]. Six undescribed 4-quinolone alkaloids, including four racemic mixtures, (±)-oxypenicinolines A–D (**66**–**69**), and two related ones, penicinolines F (**70**) and G (**71**), together with seven known analogs 1,2,3,11b-tetrahydroquinolactacide (**72**), quinolactacide (**73**), penicinoline (**74**), methyl-penicinoline (**75**), penicinoline E (**76**), quinolonimide (**77**), and 4-oxo-1,4-dihydroquinoline-3-carboxamide (**78**), were isolated from mangrove *Avicennia marina* endophytic fungus *Penicillium steckii* SCSIO 41025 (Zhanjiang, Guangdong province, China). Compounds (±)-oxypenicinoline A (**66**) and quinolactacide (**73**) displayed *α*-glucosidase inhibitory activity, with IC_50_ values of 317.8 and 365.9 μM, respectively, which were more potent than the positive-control acarbose (IC_50_, 461.0 μM). Additionally, penicinoline (**74**) and penicinoline E (**76**) showed moderate inhibitory activity against acetylcholinesterase (AChE), with IC_50_ values of 87.3 and 68.5 μM, respectively [58] (Figure 3).

#### 2.1.4. Sclerotioramines

Two new sclerotioramines sclerketides E (**79**) and F (**80**), and a new natural product sclerketide G (**81**), together with four known compounds scleratioramine (**82**), isochromophilone VI (**83**), IX (**84**), and penazaphilone G (**85**), were isolated from *Bruguiera gymnorhiza* endophytic fungus *Penicillium sclerotiorin* SCNU-F0040 (Zhanjiang province, Guangdong province, China). Compounds **79** and **80** had an *α*-glucosidase inhibitory activity with IC_50_ values of 102.3 and 217.5 μM, respectively [59]. Six new sclerotioramines, sclerazaphilones A-F (**86**–**91**), were isolated from *Hibiscus tiliaceus*-derived fungus *Penicillium sclerotiorum* ZJHJJ-18 (Zhanjiang, Guangdong province, China). Compounds **88**–**90** exhibited effective inhibitory effects on the nitric oxide (NO) production in lipopolysaccharide (LPS)-induced RAW264.7, with IC_50_ values of 7.87, 6.30, and 9.45 μM, respectively, and had no toxicity to macrophage RAW 264.7 at 50 μM [60] (Figure 4).

#### 2.1.5. Pyridine Derivatives

Two new alkaloids pyripyropenes V (**92**) and W (**93**), along with one known alkaloid pyripyropene A (**94**), were isolated from *Sonneratia paracaseolaris* rhizospheric soil derived-fungus *Penicillium robsamsonii* HNNU0006 (Dongzhai Gang, Haikou, Hainan province, China) [36]. One new compound, (4*E*)-1-(4,6-dihydroxy-5-methylpyridin-3-yl)hex-4-en-1-one (**95**), was isolated from the rhizosphere soil of *Hibiscus tiliaceus* Linn-derived fungus *Penicillium* sp. DM815 (Qinglangang, Wenchang, Hainan province, China) [47] (Figure 5).

#### 2.1.6. Benzodiazepines

A new compound, 7-hydroxy-3,10-dehydrocyclopeptine (**96**), and two known compounds, arctosin (**97**) and 3,10-dehydrocyclopeptine (**98**), were identified from the sediment of the mangrove-derived fungus *Penicillium polonicum* MCCC3A00951 (Zhangjiangkou, Fujian province, China) [46]. Two known compounds, cyclopenin (**99**) and cyclopeptine (**100**), were identified from the sediments of the mangrove-derived fungus *Penicillium polonicum* H175 (Zhangjiangkou, Fujian province, China) [48]. One known compound, 2-chlorounguinol (**101**), was identified from the mangrove *Thespesia populnea*-derived endophytic fungus *Penicillium citrinum* YX-002 (Zhanjiang, Guangdong province, China) [57] (Figure 6).

#### 2.1.7. Other Alkaloids

Two known compounds, 3-*O*-meth-ylviridicatin (**102**) and carbostyril (**103**), were identified from the sediment of the mangrove-derived fungus *Penicillium polonicum* MCCC3A00951 (Zhangjiangkou, Fujian province, China) [46]. One known compound, aspergillusidone A (**104**), was identified from the leaves of the mangrove plant *Thespesia populnea* endophytic fungus *Penicillium citrinum* YX-002 (Zhanjiang, Guangdong province, China) [57]. One known compound, chrysogine (**105**), was isolated from the rhizosphere soil of *Hibiscus tiliaceus* Linn-derived fungus *Penicillium* sp. DM815 (Qinglangang, Wenchang, Hainan province, China) [47]. One new alkaloid, penifuranone A (**106**), and two known compounds, viridicatin (**107**) and viridicatol (**108**), were isolated from the mangrove *Acanthus ilicifolius* L. endophytic fungus *Penicillium crustosum* SCNU-F0006 (Yangjiang, Guangdong province, China). Compounds **106** and **108** displayed moderate inhibitory activity against *Fusarium oxysporum* and *Penicillium italicum*, with the same MIC value of 0.25 mg/mL. Compound **107** displayed moderate inhibitory activity against *F. oxysporum* and *P. italicum*, with MIC values of 0.25 and 0.5 mg/mL, respectively. Compound **107** inhibited the growth of *Pseudomonas aeruginosa*, with an MIC value of 0.125 mg/mL. Compound **106** showed significant inhibitory activity on NO production in RAW264.7 cell lines, with an IC_50_ value of 37.01 µM, compared with the positive-control dexamethasone (IC_50_, 58.21 µM). Compound **106** showed moderate radical DPPH radicals’ scavenging activity, with an IC_50_ value of 180.2 µM [61]. One new epimer pair of long-chain polyenes, penicilqueis E (**109**) and F (**110**), were obtained from *Ceriops tagal*-derived fungus *Penicillium herquei* JX4 (South China Sea, China). Compounds **109** and **110** exhibited anti-inflammatory activity, with the IC_50_ values of 12.18 ± 0.08 μM and 10.52 ± 0.12 μM, respectively [37]. One known compound, 3-methyl-4(3*H*)-quinazolinone (**111**), was identified from mangrove sediment-derived Penicillium *polonicum* H175 (Zhangjiangkou, Fujian province, China) [48]. Three known compounds, thymine (**112**), (8*R*,9*S*)-dhydroisoflavipucine (**113**), and uridine (**114**), were isolated from mangrove sediment-derived fungus *Penicillium chrysogenum* ZZ1151 (Indonesia) [56]. Two new nitrogenous compounds, penicmariae-crucis C acid (**115**) and *N*-(6-hydroxy-2-oxoindolin-3-ylidene)-5′-methoxy-5′-oxobutyl-amine oxide (**116**), together with five known compounds, 6-hydroxyindole-3-carboxylic acid (**117**), *N*-acetyl-*β*-oxotryptamine (**118**), 6-hydroxy-*N*-acetyl-*β*-oxotryptamine (**119**), GKK1032C (**120**), and trichodermamide C (**121**), were isolated from mangrove *Avicennia marina* endophytic fungus *Penicillium steckii* SCSIO 41025 (Zhanjiang, Guangdong province, China). Compounds **117** and **119** exhibited potent *α*-glucosidase inhibitory activity with IC_50_ values of 64.1 and 191.4 μM, respectively, which were greatly stronger than that of the positive-control acarbose (IC_50_, 373.3 μM). Compound **120** exhibited potent antibacterial activity against *S. aureus*, with the MIC value of 3.9 μg/mL [62] (Figure 7).

### 2.2. Polyketides

Polyketides are a large group of secondary metabolites, characterized by their remarkable structural and functional diversity. These compounds display a broad spectrum of biological activities, including antibacterial, antifungal, anticancer, antiviral, immunosuppressive, cholesterol-lowering, and anti-inflammatory effects. A total of 146 antibacterial polyketides were obtained from mangrove ecosystem-derived *Penicillium* sp., comparing 40 lactone derivatives, 32 pyrone derivatives, 12 azaphilone derivatives, and 62 other polyketide metabolites. Notably, 61 of these compounds were novel, and 47 demonstrated biological activity.

#### 2.2.1. Lactone Derivatives

Four previously undescribed curvularin derivatives, sumalarins D–G (**122**–**125**), along with two known related compounds, curvularin (**126**) and dehydrocurvularin (**127**), were isolated and identified from the mangrove *Bruguiera sexangular* endophytic fungus *Penicillium sumatrense* MA-325 (Hainan province, China). Compounds **123** and **126** were active against aquatic pathogenic bacteria *Vibrio alginolyticus* and *V. harveyi*, with the same MIC value of 64 μg/mL. Compound **122** was active against aquatic pathogenic bacteria *V. alginolyticus* and *V. harveyi*, with MIC values of 16 and 8 μg/mL, respectively. Compound **127** showed antibacterial activity against *V. alginolyticus* and *V. harveyi*, with MIC values of 4 and 8 μg/mL, respectively. Compound **127** showed cytotoxic activity against tumor cell lines 5673, HCT 116, 786-O, and Hela, with IC_50_ values of 3.5, 10.6, 10.9, and 14.9 μM, respectively [63]. Two new isocoumarins, peniciisocoumarins I (**128**) and J (**129**), together with two known analogs, 3(*R*)-7-hydroxy-8-methoxy-3-(4-oxopentyl) isochroman-1-one (**130**) and penicimarin C (**131**), were obtained from *Penicillium* sp. GXIMD 03001, an endophytic fungus derived from the mangrove *Kandelia candel* (Beibu Gulf, Guangxi province, China) [64]. Three known compounds, aspergillumarin B (**132**), aspergilactone B (**133**), and territrem B (**134**), were isolated from the mangrove *Xylocarpus granatum* Koenig-derived fungus *Penicillium verruculosum* TGM14 (South China Sea) [44]. A new phenolic compound, 6-(2-acetyl-3,5-dihydroxybenzyl)-4-hydroxy-3-methyl-2*H*-pyran-2-one (**135**) was isolated from mangrove rhizosphere soil-derived fungus *Penicillium janthinellum* HK1-6 (Dongzhaigang, Haikou, Hainan province, China). A plausible biosynthetic route of **135** was proposed (Figure 2). The biosynthesis commences with the formation of the polyketide chain via Claisen condensation, employing one acetyl-CoA starter unit and six malonyl-CoA extender units. Subsequently, the first cyclization event involves an aldol condensation between C-7 and C-12. After aromatization of the first ring and formation of the *α*-pyrone, amethyl is attached to C-2 in the presence of *S*-adenosylmethionine (SAM) to give compound **135** [65]. Two new polyketides, penicillols A (**136**) and B (**137**), featuring spiroketal rings, along with two known analogs, dichlorodiaportal (**138**) and citreoviranol (**139**), were isolated from the mangrove *Avicennia marinav* endophytic fungus *Penicillium* sp. BJR-P2 (Yangjiang, Guangxi province, China). Compound **137** showed anti-inflammatory activity and inhibited LPS-induced NO production in RAW 264.7 cells, with an IC_50_ value of 12 µM, which was more potent than the positive-control indomethacin (IC_50_, 35.8 ± 5.7 µM) [66]. One known compound, 5,6,8-trihydroxy-4-(1′-hydro-xyethyl) isocoumarin (**140**), was isolated from endophytic fungus *Penicillium* sp. GXIMD 03101, which was collected from the mangrove *Acanthus ilicifolius* L. collected in the South China Sea [67]. Two new isocoumarins penicimarins L (**141**) and M (**142**), along with seven known analogs peniciisocoumarin E (**143**), aspergillumarin A (**144**), penicimarin I (**145**), peniciisocoumarin F (**146**), penicilloxalone B (**147**), penicimarin G (**148**), and penicimarin H (**149**), were isolated from mangrove *Xylocarpus granatum*-derived fungus *Penicillium* sp. MGP11 (Sanya, Hainan province, China). Compound **148** had better antioxidant activity (ABTS radical), with an IC_50_ value of 4.6 μM, better than positive-control trolox (IC_50_, 12.9 μM). Compounds **145**, **148**, and **149** exhibited *α*-glucosidase inhibitory activity, with IC_50_ values of 776.5, 683.7 and 868.7 μM, respectively [39]. Three new isocoumarin derivatives, (*S*)-6,8-dihydroxy-5-(methoxymethyl)-3,7-dimethylisochroman-1-one (**150**), (*S*)-6,8-dihydroxy-3,5,7-trimethyl-isochroman-1-one (**151**), and (*R*)-2-chloro-3-(8-hydroxy-6-methoxy-1-oxo-1*H*-isochromen-3-yl) propyl acetate (**152**), along with one known compound monaschromone (**153**), were isolated from mangrove plant *Heritiera littoralis*-derived fungus *Penicillium* sp. YYSJ-3 (Dongzhaigang Mangrove Nature Reserve, Guangdong province, China). Compound **152** showed promising inhibitory activity against *α*-glucosidase (IC_50_, 100.6 μM), which was stronger than the positive-control 1-deoxynojirimycin (IC_50_, 141.2 μM) [68]. One new long-chain polyene, pinophol G (**154**), was obtained from *Ceriops tagal*-derived fungus *Penicillium herquei* JX4, collected in the South China Sea. Compound **154** showed moderate anti-inflammatory activity, with the IC_50_ value of 33.26 ± 0.14 μM [37]. Two secondary metabolites, penicillic acid (**155**) and brefeldin A (**156**), were isolated from the mangrove rhizosphere soil-derived fungus *Penicillium javanicum* HK1-23 and *P. janthinellum* HK1-6, respectively (Dongzhaigang, Haikou, South China Sea). Compound **156** showed potent antifungal activity toward *Rhizoctonia solani* and *R. cerealis*, with 67.5% and 76% growth inhibition, respectively, at 50 μg/mL [69]. One known compound, 7-dehydrobrefeldin A (**157**), was isolated from mangrove sediment-derived fungus *Penicillium brefeldianum* ABC190807 (Dongzhaigang Mangrove Nature Reserve, Hainan province, China) [51]. One known compound, (+)-(5*R*,5′*R*)-3,3′-methylenebistetronic acid (**158**), was isolated from the mangrove *Acanthus ilicifolius* L. endophytic fungus *Penicillium crustosum* SCNU-F0006 (Yangjiang Mangrove Nature Reserve in Guangdong province, China). Compound **158** displayed inhibitory activity against *Fusarium oxysporum*, *Penicillium italicum*, *Colletrichum litchi*, *Pseudomonas aeruginosa*, *Salmonella typhimurium*, *Escherichia coli*, and *S. aureus*, with MIC values of 0.25, 0.25, 0.5, 0.5, 1, 1, and 1 mg/mL, respectively. Compound **158** showed significant anti-inflammatory activity (IC_50_, 37.01 µM), compared with the positive-control dexamethasone (IC_50_, 58.21 µM). Compound **158** showed moderate DPPH radical scavenging activity, with an IC_50_ value of 120.5 µM [61]. One known polyketide, astalaminoid C (**159**), was obtained from mangrove forest-derived fungus *Penicillium* sp. HDN15-312 (Sanya, Hainan province, China). Compound **159** showed DPPH radicals’ scavenging activity, with an IC_50_ value of 32.11 µM [70]. Two compounds, (+)-semivioxanthin (**160**) and (−)-striatisporolide A (**161**), were obtained from *Acanthus ilicifolius*-derived fungus *Penicillium* sp. HS-N-29, collected from the South China Sea. Compound **160** exhibited anti-angiogenic activity to *Zebrafish embryo* at 1 μg/mL, associated with notochord malformation and tail malformation (28 h) [60] (Figure 8).

#### 2.2.2. Pyrone Derivatives

One poyrone derivative, penialidin C (**162**), was isolated from mangrove sediment-derived fungus *Penicillium* sp. SCSIO 41411, which was collected from Gaoqiao mangrove wetland in Zhanjiang, coastline of the northern part of Beibu Gulf, China [71]. One known phenolic derivative, 7-hydroxy-2-(hydroxymethyl)-5-methyl-4*H*-chromen-4-one (**163**), was isolated from the mangrove rhizosphere soil-derived fungus *Penicillium janthinellum* HK1-6 (Dongzhaigang Mangrove Natural Reserve, Hainan province, China) [65]. Two polyketides, penialidins A (**164**) and C (**165**), were isolated from the mangrove rhizosphere soil-derived fungus *Penicillium javanicum* HK1-23 (Dongzhaigang, Hainan province, China). Compound **164** showed inhibitory activity against three *S. aureus* strains (ATCC 33591, ATCC 25923, ATCC 29213), with MIC values of 12.5, 25, and 6.25 μg/mL, respectively. In particular, **164** exhibited significant antibacterial activity against the methicillin-resistant *S. aureus* ATCC 43300 with an MIC value of 0.78 μg/mL, comparable to that of the positive-control, vancomycin. Compound **165** showed inhibitory activity against two *S. aureus* strains (ATCC 43300 and ATCC 29213), with MIC values of 3.13 and 12.5 μg/mL, respectively [42]. One new polyketide, talamin E (**166**), and one known polyketide, talamin B (**167**), were obtained from a mangrove-derived fungus *Penicillium* sp. HDN15-312 (Sanya, Hainan province, China). Compound **166** showed DPPH radicals’ scavenging activity, with an IC_50_ value of 6.79 µM, better than the positive-control vitamin C (IC_50_, 12.15 µM) [70]. Three new *α*-pyrone derivatives, annularins L-N (**168**–**170**), were isolated from mangrove plant *Rhizophora mucronata*-derived fungus *Penicillium herquei* MA-370 (Hainan province, China). Compound **168** displayed weak activity against *E. coli*, *Edwardsiella tarda*, *Micrococcus luteus*, *V. alginolyticus*, and *Pseudomonas aeruginosa*, with MIC values of 64, 32, 32, 64, and 64 μg/mL, respectively. Compound **169** displayed weak activity against *E. coli*, *E. tarda*, *M. luteus*, and *P. aeruginosa*, with MIC values of 64, 32, 64 and 64 μg/mL, respectively [72]. Two naphtho-*γ*-pyrones, peninaphones A (**171**) and B (**172**), were isolated from mangrove rhizosphere soil-derived fungus *Penicillium* sp. HK1-22. Compounds **171** and **172** exhibited antibacterial activity against *Enterococcus faecium* ATCC 35667, producing inhibition zones of 11.9 and 11.7 mm, respectively, and *E. faecium* ATCC 51299 with antibacterial inhibition zones of 11.2 and 16.8 mm, respectively, at the tested dosages of 25 μg per disk [40]. One known compound, trichopyrone (**173**), from mangrove rhizosphere soil of *Hibiscus tiliaceus* Linn.-derived fungus *Penicillium* sp. DM815 (Qinglangang, Wenchang, Hainan province, China), weakly inhibited LPS-induced NO release at 10 μM [47]. One new compound 6-acetyl-4-methoxy-3,5-dimethyl-2*H*-pyran-2-one (**174**), and nine known compounds 4-hydroxy-3,6-dimethyl-2*H*-pyran-2-one (**175**), verrucosidinol (**176**), methyl verrucosidinol (**177**), verrucosidinol acetate (**178**), normethylverrucosidin (**179**), verrucosidin (**180**), deoxyverrucosidin (**181**), aurantiogrisidinol (**182**), and penicyrone A (**183**), were identified from the mangrove-derived fungus *Penicillium polonicum* H17 (Zhangjiangkou, Fujian province, China) [48]. Two new citreoviridin derivatives, citreoviridins H (**184**) and I (**185**), along with one known analog citreopyrone D (**186**), were isolated from the mangrove *Avicennia marinav* endophytic fungus *Penicillium* sp. BJR-P2 (Yangjiang Hailing Island Mangrove Wetland Park, China) [66]. Six new *α*-pyrone polyketides, penipyrols C-G (**187**–**191**) and methyl-penipyrol A (**192**), together with one biogenetically related known compound, penipyrol A (**193**), were isolated from the rhizosphere soil of *Rhizophora stylosa*-derived fungus *Penicillium* sp. HDN-11-131 (Yingluo Bay, Guangxi province, China). Compounds **187**–**190** possessed a rare skeleton featuring *γ*-butyrolactone linked to *α*-pyrone ring through double bond. Compound **187** induced pancreatic *β*-cell regeneration in zebrafish at 10 μM [73] (Figure 9).

#### 2.2.3. Azaphilone Derivatives

Seven pairs of azaphilones *E*/*Z* isomers, isochromophilone H (**194a**/**194b**), sclerotiorins A–B (**195a**/**195b** and **196a**/**196b**), ochlephilone (**197a**/**197b**), isochromophilone IV (**198a**/**198b**), isochromophilone J (**199a**/**199b**), and isochromophilone I (**200a**/**200b**), were isolated from unidentified mangrove-derived fungus *Penicillium sclerotiorum* HY5 (South China Sea, Haikou, China). Compounds **196**, **197**, and **200** exhibited potent phytotoxicity against the growth of plumule on *Amaranthus retroflexus* L., with EC_50_ values of 320.84, 287.07, and 288.36 µM, respectively, and for radicle on *A. retroflexu*s L., with EC_50_ values of 271.48, 234.87, and 240.30 µM, respectively, compared to the positive-control, glufosinate-ammonium (EC_50_, 656.04 and 555.11 µM, respectively) [74]. Two known compounds, geumsanol G (**201**) and 6-((3*E*,5*E*)-5,7-dimethyl-2-methyl-enenona-3,5-dienyl)-2,4-dihydroxy-3-methylbenza-ldehyde (**202**), were isolated from the mangrove plant *Bruguiera gymnorhiza* endophytic fungus *Penicillium sclerotiorin* SCNU-F0040 (Zhanjiang Mangrove Nature Reserve, Guangdong province, China) [59]. Three new azaphilone derivatives, sclerazaphilones G–I (**203**–**205**), were isolated from *Hibiscus tiliaceus*-derived fungus *Penicillium sclerotiorum* ZJHJJ-18 (Zhanjiang Mangrove Nature Reserve, Guangdong province, China). Compounds **203** and **205** exhibited moderate anti-inflammatory activity, with IC_50_ values of 36.50 and 44.87 μM, respectively [60] (Figure 10).

#### 2.2.4. Other Polyketides

Four new polyketide decalin derivatives, penicisteck acids A–D (**206**–**209**), and two known compounds, tanzawaic acids B (**210**) and Z (**211**), were obtained from mangrove *Avicennia marina* endophytic fungus *Penicillium steckii* SCSIO 41025 (Zhanjiang, Guangdong province, China). A plausible biosynthetic pathway for compounds **206**–**211** has been proposed, suggesting that these compounds may originate from successive Claisen condensations involving an acetyl coenzyme A (CoA) and five malonyl-CoA molecules (Figure 3) [62]. Eighteen new compounds, penicisteck acid I (**212**), *epi*-penitanzacid G (**213**), penicisteck acids J (**214**) and K (**215**), penicisteck acids L-Y (**216**–**229**), and sixteen known compounds, arohynapenes A (**230**) and B (**231**), tanzawaic acid A (**233**), tanzawaic acids X (**232**), D (**234**), N (**235**), O (**236**), V (**237**), C (**238**), E (**239**), M (**240**), Z1 (**242**), S (**243**), H (**244**), 10-hydroxytanzawaic acid T (**241**), penitanzacid H (**245**) and 12*Z*,14*E*-tanzawaic acid D (**246**), were isolated from mangrove plant of *Avicennia marina* endophytic fungus *Penicillium steckii* SCSIO 41025 (Zhanjiang, Guangdong province, China). Compounds **213**, **218**, **220**–**221**, **223**–**226**, **229**, **236**, **237**, **238**, **243**, and **245** suppressed LPS-induced NF-*κ*B luciferase in a dose-dependent manner, with IC_50_ values of 13.67, 21.81, 26.29, 19.84, 12.92, 12.84, 14.39, 20.55, 17.95, 14.04, 12.37, 14.61, 18.70, and 16.74 μM, respectively [75]. One polyketide derivative, brefeldin G (**247**), was isolated from mangrove sediment-derived fungus *Penicillium* sp. SCSIO 41411 (Gaoqiao mangrove wetland in Zhanjiang, Beibu Gulf, China) [71]. Three new polyketides, furantides A–B (**248**–**249**) and arugosinacid A (**250**), were obtained from mangrove-derived fungus *Penicillium* sp. HDN15-312 (Sanya, Hainan province, China). Compound **250** had DPPH radical eliminating activity, with an IC_50_ value of 56.92 µM [70]. One known compound, terrein (**251**), was isolated from the mangrove *Xylocarpus granatum* Koenig-derived fungus *Penicillium verruculosum* TGM14, and mangrove plant (South China Sea, China) [38]. One known benzopyran derivative, *cis*-(3*R*,4*S*)-3,4-dihydro-3,4,8-trihydroxynaphthalen-1(2*H*)-one (**252**), was obtained from mangrove *Kandelia candel*-derived fungus *Penicillium citrinum* QJF-22 (Haikou East Harbour National Nature Reserve, Hainan province, China). Compound **252** exhibited moderate anti-inflammatory activity, with an IC_50_ value of 44.7 μM, and without cytotoxicity to RAW264.7 cells within 50 μM [35]. One new xanthene derivative, penicixanthene E (**253**), was isolated from the mangrove *Acanthus ilicifolius* L. endophytic fungus *Penicillium* sp. GXIMD 03101 (South China Sea). Compound **253** is a xanthene derivative that was initially reported with its carbon–carbon double bond reduced, which exhibited cytotoxic activity against pancreatic cancer SW1990, with an IC_50_ value of 23.8 μM [58]. One known compound, *trans*-3,4-dihydro-3,4,8-trihydroxynaphthalen-1(2*H*)-one (**254**), was identified from the mangrove sediment-derived fungus *Penicillium polonicum* H175 (Zhangjiangkou, Fujian province, China) [48]. Five known compounds, 10,11,16,17-tetrahydrobislongiquinolide (**255**), epitetrahydrotrichodimer ether (**256**), (9′*R*)-tetrahydrotricho-dimer ether (**257**), trichobisvertinol A (**258**), and dihydrotrichodermolide (**259**), were isolated from mangrove rhizosphere soil of *Hibiscus tiliaceus* Linn.-derived fungus *Penicillium* sp. DM815 (Qinglangang, Wenchang, Hainan province, China). Compounds **256** and **257** inhibited LPS-induced NO release at 10 μM [47]. Four new compounds, aceneoherqueinones A (**260**) and B (**261**), (+)-aceatrovenetinones A (**262**) and B (**263**), along with four known congeners, (–)-aceatrovenetinones A (**264**) and B (**265**), (–)-scleroderolide (**266**), and (+)-scleroderolide (**267**), were isolated from *Rhizophora mucronata*-derived fungus *Penicillium herquei* MA-370 (Hainan province, China). Compounds **260** and **261** displayed inhibitory activity against angiotensin converting enzyme (ACE), with IC_50_ values of 3.10 and 11.28 μM, respectively [76] (Figure 11).

### 2.3. Terpenoids

Terpenoids constitute a significant class of natural compounds, characterized by their chemical structural diversity and substantial biological activity. These compounds are produced by a wide array of plant and fungal genera. A total of 84 terpenoids (including 48 new compounds, and 26 compounds with bioactivity) were found, comprising 10 sesquiterpenes and 74 meroterpenoids.

#### 2.3.1. Sesquiterpene

One new cadinane-type sesquiterpene (3*β*,4*β*,5*β*,6*β*,7*β*,9*β*,10*α*)-4,6-epoxy-7-hydroxy-9-cadinanol (**268**) was isolated from the surface of mangrove leaves *Penicillium oxalicum* KMM 4683 (South China Sea) [77]. A new eudesmane-type sesquiterpenoid, artemihedinic acid N (**269**), was isolated from mangrove rhizosphere soil-derived fungus *Penicillium* sp. HK1-22 [40]. Five new sesquiterpenoids, citreobenzofurans D-F (**270**–**272**) and phomenones A–B (**273**–**274**), along with one known compound xylarenone A (**275**), were isolated from mangrove root soil-derived fungi *Penicillium* sp. HDN13-494 (Wenchang, Hainan, China). Compound **274** showed moderate activity against *B. subtilis*, with an MIC value of 6.25 µM [78]. One new compound astellolide Q (**276**), and a known compound, compound V (**277**), were isolated from rhizosphere soil of mangrove *Avicennia marina*-derived fungus *Penicillium* sp. N-5 (Nansha Mangrove National Nature Reserve, Guangdong province, China). Compound **276** exhibited antimicrobial activity against *P. italicum* and *C. Gloeosporioides*, with the same MIC value of 25 µg/mL [79] (Figure 12).

#### 2.3.2. Meroterpenoids

Three unusual austin-type meroterpenoids, penicianstinoids C-E (**278**–**280**), were obtained from the mangrove *Brguiera sexangula*-derived fungus *Penicillium* sp. TGM112 (South China Sea). Compounds **278** and **280** inhibited the growth of newly hatched *Helicoverpa armigera larvae*, with IC_50_ values of 100 and 200 μg/mL, respectively. A plausible biosynthetic pathway for compounds **278**–**280** has been proposed, suggesting that these compounds may originate from the incorporation of 3,5-dimethylorsellinic acid with farnesyl pyrophosphate, followed by a series of reactions (Figure 4) [32]. Eight austin meroterpenoids, preaustinoids A2 (**281**) and D (**282**), 11*β*-acetoxyisoaustinone (**283**), austinolide (**284**), austinol (**285**), austin (**286**), dehydroaustinol (**287**), and acetoxydehydroaustin (**288**), were isolated from the root of mangrove *Xylocarpus granatum*-derived fungus *Penicillium* sp. MGP11 (Sanya Tielugang Mangrove Nature Reserve, Hainan province, China). Compounds **282**–**283** and **286**–**288** showed strong inhibitory activity against *α*-glucosidase, with the IC_50_ values of 249.5, 224.5, 252.3, 270.4, and 289.7 μM, respectively, which were stronger than the positive-control acarbose (IC_50_, 313.9 μM) [41]. Fifteen new meroterpenoids, littoreanoids A–O (**289**–**303**), including three rearranged skeleton meroterpenoids (**289**–**291**), were isolated from the root of mangrove *Lumnitzera littoreav*-derived fungus *Penicillium* sp. HLLG-122 (Sanya, Hainan province, China). Compound **289** was a novel berkeleyacetal-derived meroterpenoid featuring an unusual spirocyclic 2-oxaspiro[5.5]undeca-4,7-dien-3-one moiety. Compound **290** possessed an unusual 6/6/6/6/6 pentacyclic system with a novel 1-hydroxy-7,7-dimethyl-2-oxabicyclo[2.2.2]octan-5-yl acetate moiety. Compound **291** was an unusual 6/7/6/5/6/5/4 polycyclic system containing a *β*-lactone ring. Compounds **294** and **299** exhibited anti-inflammatory effects, with IC_50_ values of 30.41 and 19.02 μM, respectively. Compound **299** could suppress the levels of TNF-*α* and IL-6, and downregulate the protein expression of iNOS and COX-2 in RAW 264.7 cells. A plausible biosynthetic pathway for compounds **289–291** has been proposed, suggesting that they may originate from 3,5-dimethylorsellinic acid and the terpenoid precursor farnesyl diphosphate, following a series of reactions (Figure 5) [80]. Three new andrastin-type meroterpenoids hemiacetalmeroterpenoids A–C (**304**–**306**), together with ten known compounds (3-deacetyl-citreohybridonol (**307**), citreohybridone A (**308**), 3,5-dimethylorsellinic acid-based meroterpenoid 2 (**309**), andrastins A–C (**310**–**312**), andrastone C (**313**), penimeroterpenoid A (**314**), 23-deoxocitreohybridonol (**315**), and 6*α*-hydroxyandrastin B (**316**)) were isolated from the rhizosphere soil of mangrove *Avicennia marina*-derived fungus *Penicillium* sp. N-5 (Nansha Mangrove National Nature Reserve, Guangdong province, China). Hemiacetalmeroterpenoid A (**304**) was a new andrastin-type meroterpenoid containing a unique 6,6,6,6,5,5-hexa-cyclic skeleton. Compound **304** exhibited remarkable antimicrobial activity against *B. Subtilis*, *P. italicum*, and *C. gloeosporioides*, with the same MIC value of 6.25 µg/mL. Compound **308** exhibited potent antimicrobial activity against *P. italicum* and *C. gloeosporioides*, with MIC values of 1.56 and 3.13 µg/mL, respectively. Compound **309** exhibited antimicrobial activity against *P. italicum* and *C. gloeosporioides*, with MIC values of 12.5 and 25 µg/mL, respectively. Compound **311** exhibited strong antimicrobial activity against methicillin-resistent *S. aureus*, *B. subtilis*, *P. aeruginos*a, *S. typhimurium*, *P. italicum*, and *C. gloeosporioides*, with MIC values of 25, 12.5, 25, 3.13, 6.25, and 6.25 µg/mL, respectively [79]. Based on the guidance of molecular networking and OSMAC approach, nine new highly oxygenated meroterpenoids, peniciacetals A–I (**317**–**325**), along with five known analogs, berkeleyacetal A (**326**) and chrysogenolides B–E (**327**–**330**), were isolated from the root of mangrove *Lumnitzera littorea-derived* fungus *Penicillium* sp. HLLG-122 (Sanya, Hainan Island, China). Peniciacetals A (**317**) and B (**318**) were characterized with a unique 6/6/6/6/5 pentacyclic system, possessing an unusual 4,6-dimethyl-2,5-dioxohexahydro-6-carboxy-4*H*-furo[2,3-b]pyran moiety. Peniciacetals C (**322**) and D (**320**) possessed an uncommon 3,6-dimethyldihydro-4*H*-furo[2,3-b]pyran-2,5-dione unit with a 6/6/6/5/6 fused pentacyclic skeleton. Compound **330** showed good cytotoxicity against HepG2, MCF-7, HL-60, A549, HCT116, and H929 cell lines, with IC_50_ values of 6.6, 14.8, 3.2, 5.7, 6.9, and 3.0 μM, respectively. The plausible biosynthetic pathway of compounds **317**–**325** were also proposed, these compounds are formed by farnesyl pyrophosphate and 3,5-dimethylorsellinic acid (Figure 6) [81]. One known compound, citreohybridonol (**331**), was isolated from mangrove rhizosphere soil of *Hibiscus tiliaceus* Linn-derived fungus *Penicillium* sp. DM815 (Qinglangang, Wenchang, Hainan province, China) [47]. Ten previously undescribed meroterpenoids, cyclohexenoneterpenes A–J (**332**–**341**), together with ten known analogs peniginsengin A (**342**), penicyclone E (**343**), 22-deacetyl-yanuthone A (**344**), bisorbicillinol (**345**), 22-deacetylyanuthone A (**346**), penicillone A (**347**), penicyclone A (**348**), 4′-oxomacrophrin A (**349**), macrophorin A (**350**), and purpurogemutantin (**351**), were isolated from the rhizosphere soil of mangrove *Avicennia marina*-associated fungus *Penicillium* sp. N-5 (Nansha Mangrove National Nature Reserve, Guangdong province, China). Compounds **341**, **344**, and **345** exhibited significant cytotoxic activities against HCT-116 cells, with IC_50_ values of 8.7, 8.2, and 2.6 μM, respectively. Compound **349** had the strongest cytotoxic activity against MDA-MB-435 and HCT-116 cell lines, with the same IC_50_ value of 1.4 μM. A plausible biosynthetic pathway for compound **336** has also been proposed. It involves the preparation of a single polyketone chain (I-1) from one acetyl-CoA and two malonyl-CoAs, a process catalyzed by ketosynthase (KS) and acyltransferase (AT). Under the action of transmethylase (MT), the intermediate I-1 undergoes aldol condensation to form a 2-methylphenol (intermediate I-2). Intermediate I-3 was produced by enzymaticreaction of I-2 with farnesyl pyrophosphate. Finally, compound **336** may be produced via a series of oxidation, addition, and reduction reactions of intermediates I-3, I-4, and I-5 (Figure 7) [82] (Figure 13).

### 2.4. Benzene Derivatives

Benzene derivatives exhibit a broad range of applications and are extensively utilized within the realm of organic chemistry. A total of 38 benzene compounds (including 9 new compounds, and 15 compounds with bioactivity) were discovered from the genus *Penicillium* sp.

Three new hydroxyphenylacetic acid derivatives, stachylines H–J (**352**–**354**), together with four known compounds, stachylines G (**355**) and F (**356**), (*E*)-4-(4-hydroxy-3-methylbut-2-enyloxy)benzaldehyde (**357**), and stachyline E (**358**), were isolated from mangrove sediment-derived fungus *Penicillium* sp. SCSIO 41411 (Gaoqiao mangrove wetland in Zhanjiang, Beibu Gulf, China). Compounds **352** and **353** displayed weak inhibition activity against AChE with inhibitory ratios of 22.3% and 19.9%, respectively, at a concentration of 50 µM [71]. Five known phenolic derivatives, 3,5-dihydro-xy-2-(2-(2-hydroxy-6-methylphenyl)-2-oxoethyl)-4-methylbenzaldehyde (**359**), 3-hydroxy-5-methylphenyl 2,4-dihydroxy-6-methylben-zoate (**360**), lecanoric acid (**361**), orsellinic acid (**362**), and orcinol (**363**), were isolated from mangrove rhizosphere soil-derived fungus *P. janthinellum* HK1-6 (Dongzhaigang Mangrove Natural Reserve, Hainan province, China) [65]. Eight compounds, including a novel tetrasubstituted benzene derivative peniprenylphenol A (**364**), and seven known compounds 1,2-seco-trypacidin (**365**), communol G (**366**), clavatol (**367**), 4-hydroxybenzeneacetic acid methyl ester (**368**), 2,5-dihydroxyphenylacetic acid methyl ester (**369**), 2-hydroxyphenylacetic acid methyl ester (**370**), and 4-hydroxyphenylethanone (**371**), were isolated from mangrove sediment-derived fungus *Penicillium chrysogenum* ZZ1151 (Indonesia). Compound **366** and **371** had antibacterial activity against MRSA, with MIC value of 25 μg/mL. Compound **369** showed antifungal activity against *C. albicans*, with an MIC value of 25 μg/mL [56]. One known compound, 2′,3′-dihydrosorbicillin (**372**), was isolated from mangrove rhizosphere soil of *Hibiscus tiliaceus* Linn.-derived fungus *Penicillium* sp. DM815 (Qinglangang, Wenchang, Hainan province, China) [47]. Two known compounds, 4-((*S*)-2-hydroxybut-3-ynyloxy)-benzoic acid (**373**) and penipratynolene (**374**), were identified from sediments of the mangrove forest-derived fungus *Penicillium polonicum* H175 (Zhangjiangkou Fujian province, China) [48]. Three grisephenone-type xanthone derivatives, sephenone A (**375**), 5,9,11-trimethoxy-3,13-dihydroxybenzophenone (**376**), stachybogrisephenone B (**377**), and a diphenyl ether derivative 4-chloro-7,4′-dihydroxy-5,2′-dimethoxy-2-methylformate-6′-methybenzophone (**378**), were isolated from the rhizospheric soil of the mangrove *Sonneratia paracaseolaris*-derived fungus *Penicillium robsamsonii* HNNU0006 (Dongzhai Gang Harbour Mangrove Natural Reserve Area, Hainan province, China). Compounds **375**–**377** showed moderate cytotoxicity against tumor cell lines A549, with IC_50_ values of 9.71 ± 0.34, 9.66 ± 0.17, and 5.68 ± 0.21 μg/mL, respectively, better than the positive-control VP16 (etoposide) (IC_50_, 5.8 ± 0.23 μg/mL) [36]. One new benzophenone derivative penibenzophenone C (**379**), and a new benzophenone natural product penibenzophenone D (**380**), together with two known compounds, sulochrin (**381**) and hydoxysulochrin (**382**), were isolated from *Acanthus ilicifolius* L. endophytic fungus *Penicillium* sp. (South China Sea). Compounds **379** and **380** showed moderate antibacterial activity against MRSA with MIC values of 3.12 and 6.25 μg/mL, respectively [83]. Three known compounds, cytosporone B (**383**), dothiorelone C (**384**), and cytosporone A (**385**), were isolated from the stem of the mangrove plant *Heritiera littoralis*-derived fungus *Penicillium* sp. YYSJ-3 (Zhuhai Mangrove Nature Reserve, Guangdong province, China). Compounds **384** and **385** showed inhibitory activity against *α*-glucosidase, with IC_50_ values of 133.4 and 130.9 μM, respectively, compared to the positive-control 1-deoxynojirimycin (IC_50_, 141.2 μM) [68]. Four new compounds, penicisteck acids E–H (**386**–**389**), were isolated from mangrove *Avicennia marina* endophytic fungus *Penicillium steckii* SCSIO 41025 (Zhanjiang, Guangdong province, China). Compounds **387**–**389** suppressed LPS-induced NF-*κ*B luciferase in a dose-dependent manner, with IC_50_ values of 12.54, 12.36, and 13.83 μM, respectively [75] (Figure 14).

### 2.5. Steroids

Steroids are intricate lipophilic molecules that play a multifaceted role in the body, regulating cellular, tissue, and organ functions throughout the entire lifespan. A total of eight steroid compounds, comprising one novel compound, were identified from the mangrove-derived fungus genus *Penicillium* sp.

Five sterols, ergosterol (**390**), ganodermaside A (**391**), 5*α*,8*α*-epidioxyergosta-6,22-dien-3*β*-ol (**392**), (3*β*,5*α*,6*β*,22*E*)-ergosta-7,22-diene-3,5,6-triol (**393**), and (20*S*,22*E*,24*R*)-ergosta-7,22-diene-3*β*,5*α*,6*β*,9-tetraol (**394**), were isolated from the mangrove *Xylocarpus granatum* Koenig-derived fungus *Penicillium verruculosum* TGM14 (South China Sea) [42]. One new purinyl-steroid, ergosta-4,6,8(14),22-tetraen-3-(6-amino-9*H*-purin-9-yl) (**395**), along with two known compounds, (22*E*)-ergosta-4,6,8(14),22-tetraen-3-one (**396**) and ergosterol peroxide (**397**), were isolated from mangrove sediment-derived fungus *Penicillium brefeldianum* ABC190807 (Dongzhaigang Mangrove Nature Reserve, Hainan province, China) [51] (Figure 15).

### 2.6. Other Classes

Furthermore, additional secondary metabolites have been isolated from mangrove-derived *Penicillium* sp., including fatty acids. A total of 20 compounds were obtained, with 8 being novel and 6 displaying bioactivity.

Four known compounds, 9,12-octadecadieonic acid (**398**), *α*-linolenic acid (**399**), linoleic acid (**400**), and glycerol monlinoleate (**401**), were isolated from mangrove sediment-derived fungus *Penicillium* sp. SCSIO 41411 (Gaoqiao mangrove wetland in Zhanjiang, Beibu Gulf, China) [71]. One known compound, verrucosal (**402**), was isolated from mangrove sediment-derived fungus (Zhangjiangkou in Fujian province, China) [39]. A new compound, penicillquei C (**403**), and two known compounds, talaromydien A (**404**) and penicillquei A (**405**), were isolated from the mangrove *Xylocarpus granatum* Koenig-derived fungus *Penicillium verruculosum* TGM14 (South China Sea) [38]. A new aliphatic compound, (*S*)-(3*E*,5*Z*,10*E*)-8-hydroxy-trideca-3,5,10,12-tetraen-2-one (**406**), was obtained from a mangrove *Kandelia candel* endophytic fungus *Penicillium citrinum* QJF-22 (Haikou East Harbour National Nature Reserve, Hainan province, China) [35]. One new nitrogenous compound, methyl-1′-(*N*-hydroxyacetamido)butanoate (**407**), was isolated from mangrove *Avicennia marina* endophytic fungus *Penicillium steckii* SCSIO 41025 (Zhanjiang, Guangdong province, China) [62]. One known compound, penicillar E (**408**), was obtained from *Ceriops tagal*-derived fungus *Penicillium herquei* JX4. Compound **408** showed moderate anti-inflammatory activity, with the IC_50_ value of 38.86 ± 0.06 μM [37]. One new compound, (2*E*,4*E*)-5-((2*S*,3*S*,4*R*,5*R*)-3,4-dihydroxy-2,4,5-trimethyl-tetrahydrofuran-2-yl)-2,4-dimethylpenta-2,4-dienal (**409**), and two known compounds, aspterric acid (**410**) and 3,4-dihydroxybenzaldehyde (**411**), were identified from the mangrove-derived fungus *Penicillium polonicum* H175 (Zhangjiangkou national nature reserve, Fujian province, China). Compound **413** exhibited excellent hypoglycaemic effect, which is equivalent to the positive drug rosiglitazone (RSG) at 10 μM [48]. One known compound, penicimumide (**412**), isolated from mangrove sediment-derived fungus *Penicillium chrysogenum* ZZ1151 (Indonesia), exhibited antibacterial activity against *E. coli*, with an MIC value of 13 μg/mL [56]. Four new alkane derivatives, 2-methyl-3-(5-oxohexyl) maleic acid (**413**), 2-(4-hydroxyhexyl)-3-methylmaleic acid (**414**), 3-(ethoxycarbonyl)-2-methylenenonanoic acid (**415**), and 7-hydroxy-3-(methoxycarbonyl)-2-methylenenonanoic acid (**416**), were isolated from the mangrove sediment-derived fungus *Penicillium ludwigii* SCSIO 41408 (Hongsha River estuary to South China Sea, Sanya, Hainan province, China). Compound **415** showed a suppressed Receptor Activator for Nuclear Factor-*κ*B Ligand (RANKL)-induced osteoclast differentiation in bone marrow mononuclear cells (BMMCs) at 10 µM [49]. One known compound, citreoviral (**417**), was isolated from the mangrove *Avicennia marinav* endophytic fungus *Penicillium* sp. BJR-P2 (Yangjiang Hailing Island Mangrove Wetland Park, China) [66] (Figure 16).

## 3. Comprehensive Overview and Conclusions

Fungi, particularly those inhabiting specialized eco-environments, have been proven as promising and prolific sources of bioactive secondary metabolites with potent bioactivity applications [84]. Mangrove ecosystems, as unique habitats, offer favorable conditions for the colonization of fungi. These fungi are able to produce natural compounds with novel structures and significant activity, even under high salinity, low pH, anoxic conditions, and the stress of periodic tidal inundation [47,82,85]. In continuation of our work [43], we undertook a comprehensive study that concentrated on compounds derived from *Penicillium* fungi originating from mangrove ecosystems, as detailed in Table 1. The structural diversities of the secondary metabolites isolated from *Penicillium* spp. are shown in Figure 17. The compounds were categorized primarily by their structural classifications, which included alkaloids, polyketides, terpenoids, benzene derivatives, steroids, and various other compounds. Among these compounds, eleven possess novel structural skeletons. Penicianstinoid C (**278**) was the first austin-type meroterpenoid, with a unique 6/6/6/5 rearranged tetracyclic skeleton possessing two unusual spirocyclicmoieties (2-oxaspiro[5.5]undeca-4,7-dien-3-one and 6-methylene-2-oxaspiro[4.5]decane-1,4-dione). Penicianstinoid D (**279**) was an unusual austin-type meroterpenoid, featuring a 6/6/6/6 tetracyclic skeleton that incorporates an octahydro-2*H*-chromen-2-one unit. Penicianstinoid E (**280**) features a 6/5/6/6/6/5 fused hexacyclic skeleton, characterized by the rare presence of a five-membered ether ring system [32]. Penipyrols C–F (**187**–**190**) possessed a rare skeleton featuring *γ*-butyrolactone linked to *α*-pyrone ring through double bond [66]. Peniciacetals A (**317**) and B (**318**) were characterized with a unique 6/6/6/6/5 pentacyclic system, possessing an unusual 4,6-dimethyl-2,5-dioxohexahydro-6-carboxy-4H-furo[2,3-b]pyran moiety. Peniciacetals C (**319**) and D (**320**) possessed an uncommon 3,6-dimethyldihydro-4*H*-furo[2,3-b]pyran-2,5-dione unit with a 6/6/6/5/6 fused pentacyclic skeleton [81]. These findings collectively suggest that mangrove-derived *Penicillium* fungi have the potential to produce secondary metabolites featuring novel structures.

Furthermore, in mangrove ecosystems, a special ecosystem characterized by high salinity, 26 secondary metabolites from mangrove-derived *Penicillium* species were found to contain halogen atoms, such as compounds **16**–**18**, **79**–**85**, **90**–**91**, **194**–**196**, **198**–**201**, **204**–**205**, **223**–**224**, **377**–**378** and **375**. Some of them are new compounds exhibiting a range of biological activities. For example, compounds **79** and **80** had *α*-glucosidase inhibition activity with IC_50_ values of 102.3 and 217.5 μM, respectively [59]. Compound **90** exhibited anti-inflammatory activity, with an IC_50_ value of 9.45 μM [60]. Compounds **223** and **224** suppressed LPS-induced NF-*κ*B luciferase in a dose-dependent manner, with IC_50_ values of 12.92 and 12.84 μM, respectively [75]. Compounds **375** and **377** showed moderate cytotoxicity against tumor cell lines A549, with IC_50_ values of 9.71 ± 0.34 and 5.68 ± 0.21 μg/mL, respectively [36]. Nevertheless, halogen-containing compounds are relatively rare in the secondary metabolites of *Penicillium*, as compared to those from terrestrial organisms and other resources [86]. The isolated *Penicillium* species from mangrove ecosystems can serve as an important source of halogen-containing compounds.

It is worth noting that nearly 31% (126 compounds) showed biological activities, including cytotoxic, antifungal, antibacterial, anti-inflammatory, *α*-glucosidase inhibitory activities (Figure 18). Notably, alkaloids, polyketides, and terpenoids showed a broad-spectrum biological activity. For example, dithiodiketopiperazine derivative DC1149B (**18**) exhibited inhibitory activity for *E. rhusiopathiae* WH13013 and *S. suis* SC19, with the same MIC value of 50 μg/mL. In addition, **18** showed cytotoxicity against prostate cancer cell lines 22Rv1 and PC-3, with IC_50_ values of 13.6 and 5.1 µM, respectively [49]. Lactone derivative penicillol B (**137**) showed anti-inflammatory activity, which inhibited lipopolysaccharide (LPS)-induced NO production in RAW 264.7 cells, with an IC_50_ value of 12 µM, better than the positive-control indomethacin (IC_50_, 35.8 ± 5.7 µM) [66]. Five austin meroterpenoids, preaustinoid D (**282**), 11*β*-acetoxyisoaustinone (**283**), austin (**286**), dehydroaustinol (**287**), and acetoxydehydroaustin (**288**), showed strong inhibitory activity against *α*-glucosidase, with IC_50_ values of 249.5, 224.5, 252.3, 270.4, and 289.7 μM, respectively, which are stronger than the positive-control acarbose (IC_50_, 313.9 μM) [41].

Moreover, the *Penicillium* fungi were obtained from various mangrove ecosystems: 49% from mangrove plants, 30% from mangrove rhizosphere soil, and 21% from mangrove sediments (Figure 19). Additionally, it was discovered that 96% of the fungi originated from Hainan, China, with the remaining specimens gathered from Thailand and Indonesia.

In summary, *Penicillium* species originating from mangrove ecosystems have been established as a significant source of novel compounds and a diverse array of secondary metabolites exhibiting a broad spectrum of biological activities, revealing their great untapped potential in medicinal and agrochemical applications. As research into natural products has intensified, numerous known compounds have been repeatedly isolated and reported, making the discovery of new compounds increasingly challenging. The active compounds have only been studied at the in vitro level, with a lack of thorough investigation into their in vivo efficacy and the mechanisms of action. In this study, the integration of OSMAC with molecular networking emerges as a valuable approach for the discovery of novel bioactive substances, effectively preventing the duplication of known compounds.

In future research, molecular biology techniques, such as genome mining and heterologous expression should be applied to modulate fungal metabolism and then to targeted separation based on molecular networking guidance. We should also focus on the mechanism of action of active compounds to obtain active lead compounds. This strategy will facilitate the extraction of effective lead compounds, supporting advancements in medicine and agriculture, and ultimately contributing to the safeguarding of human life and health.

## Data Availability

All data are obtained from the references.

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
