# Peer review of "Secondary Metabolites from the Mangrove Ecosystem-Derived Fungi Penicillium spp.: Chemical Diversity and Biological Activity"

_marinedrugs, 2024, doi:10.3390/md23010007_

Round 1
Reviewer 1 Report
Comments and Suggestions for Authors
The aim of this review is the conclusions about bioactive compounds from mangrove-derived Penicillium species. The work deserves interest, but I have a number of comments, first of all, on the Introduction.
There is no justification for the novelty of this review in the Introduction. To date, several fairly new reviews [doi.org/10.1039/D1NP00041A; doi: 10.3390/md22080372; doi.org/10.3390/bioengineering9120776; doi.org/10.21577/0103-5053.20240032; doi.org/10.1080/07388551.2022.2033682; ] have been published. What does this review bring to this area?
It is unclear what time period the authors analyzed. "The last five years" is something incomprehensible and vague.
According to the authors, mangrove-associated fungi of the genus Penicillium are isolated only in the South China Sea. Is it so? What is the reason for this? Do mangroves grow only there? This needs to be commented on.
The introduction is very long and unclear. Each of the issues is covered in a very crumpled manner. The authors have paid very little attention to why mangrove inhabitants can help in solving the questions asked in the first paragraph.
Currently, it is believed that the fungi of mangrove habitats cannot be considered marine. The authors need to think about it and comment on it.
Why Penicillium species? Are they predominant among mangrove-derived fungi?
In the discussion, the authors need to analyze how interaction with mangroves affects the spectrum of secondary metabolites of fungi. Are there any special features compared to other endophytes or associates of invertebrates or terrestrial soils?
Reviewer 2 Report
Comments and Suggestions for Authors
This is a highly descriptive review about the bioactive natural products isolated from the mangrove ecosystem-derived fungi Penicillium, containing the description of 420 natural products. These natural products have been classified in six main groups according to their biosynthetic origin and a brief information about their biological activities were included. In my opinion the present manuscript could be suitable for Marine Drugs but some important issues require major revision before final acceptance. These issues are the following:
1) The main concern is that some reviews about natural products isolated from marine-derived Penicillium fungi have been recently described and there is a clear overlap between the submitted manuscript and the reported reviews. Examples of the aforementioned reviews are the following:
a) Novel bioactive natural products from marine-derived Penicillium fungi: A review (2021-2023): Marine Drugs, 2024, 22, 191
b) Natural products from mangrove sediments-derived microbes: Structural diversity, bioactivities, biosynthesis and total synthesis: Eur. J. Med. Chem. 2022, 230, 114117
c) Cytotoxic compounds from Marine fungi: Sources, structures and bioactivity: Mar. Drugs, 2024, 22, 70
In this sense, the authors should include in Introduction section a comment about published reviews, such as the previous examples, justify the novelty and differences that provides the present one with respect to those and be cited in references.
2) The presentation and drawing style of the molecular structures are very important issues on this kind of papers. In this sense, the authors must revise the following aspects:
- The drawings of some compounds must be significantly improved. In particular, compounds 1, 16, 17, 25-29, 37-58, 123, 159, 197-202, 258-260, 307-309, 401-404.
- For many compounds, stereochemistry of some chiral centres must be indicated. In particular, in compounds 13, 15, 16, 17, 107, 108, 117, 157, 187, 188, 417-419
3) I suggest that some information about the molecular structures was included, indicating if the depicted molecular structures have been confirmed (by X-ray analysis or by synthesis), or if the molecular structures have been tentatively proposed after spectroscopic analyses.
4) Revise the format of the references: Include dots after abbreviations of the name of the journal; number of volume in itallics, etc...
Round 2
Reviewer 1 Report
Comments and Suggestions for Authors
The authors took into account all my comments and corrected the manuscript. The revised manuscript can now be accepted for publication.
Author Response
Thanks very much for revising our manuscript.
Reviewer 2 Report
Comments and Suggestions for Authors
The manuscript has been revised according to my comments and suggestions, and can be accepted for publication in present form.
Author Response

(The authors gave the same response as above.)
